# Detection of the Xanthi Chryso-like Virus in New Geographical Area and a Novel Arthropod Carrier

**DOI:** 10.3390/tropicalmed8040225

**Published:** 2023-04-13

**Authors:** Marko Jankovic, Valentina Cirkovic, Gorana Stamenkovic, Ana Loncar, Marija Todorovic, Maja Stanojevic, Marina Siljic

**Affiliations:** 1Faculty of Medicine, Institute of Microbiology and Immunology, Department of Virology, University of Belgrade, 1 Dr Subotića Starijeg Street, 11000 Belgrade, Serbia; 2Group for Medical Entomology, Centre of Excellence for Food and Vector Borne Zoonoses, Institute for Medical Research, University of Belgrade, 11000 Belgrade, Serbia; 3Department for Genetic Research, Institute for Biological Research “Siniša Stanković”—National Institute of Republic of Serbia, University of Belgrade, 11000 Belgrade, Serbia; 4Institute for Biocides and Medical Ecology, 11000 Belgrade, Serbia

**Keywords:** *Culex pipiens*, mosquito, arthropod-borne, RNA virus, mycovirus, chrysovirus, sequencing, Serbia

## Abstract

Here, we report on a serendipitous finding of a chryso-like virus associated with *Culex pipiens* mosquitos in the course of study aimed to detect and characterize West Nile virus (WNV) circulating in mosquitos in Serbia, Southern Europe. Upon initial detection of unexpected product in a PCR protocol for partial WNV *NS5* gene amplification, further confirmation and identification was obtained through additional PCR and Sanger sequencing experiments. Bioinformatic and phylogenetic analysis identified the obtained sequences as Xanthi chryso-like virus (XCLV). The finding is particular for the fact that it associates XCLV with a new potential vector species and documents a novel geographical area of its distribution.

## 1. Introduction

Mosquitoes are ubiquitous insects known to harbor a large variety of RNA viruses with a significant impact on human health. Increasing use of next-generation sequencing provided insight into mosquito virome, potentiating their role as vectors of human arboviruses and highlighting a panoply of the viral species pertaining to vertebrates, invertebrates, plants, fungi, protozoa. A rich viral community of mosquito virome may facilitate genetic and other interactions (such as recombinations, reassortments) among diverse viral populations, consequences of which are difficult to predict and possibly give rise to new and emerging agents of disease [1].

Viruses infecting fungi have been described in many fungal species, including medically important ones such as *Aspergillus fumigatus* [2,3,4,5,6]. For some of these fungal viruses, they are hypothesized to be directly and/or indirectly favorable to humans in that they interfere with fungal physiology, such as the mycoviruses of *Alternaria alternata* and the pervasive rice blast fungus *Magnaporthe oryzae* [7,8,9,10]. In addition, a possible role for fungal viruses as agents of biological pest control has been suggested [11]. Chrysoviruses are among the recognized mycopathogens so far shown to be associated with mosquitoes [12,13].

*Chrysoviridae* are small, multipartite, double-stranded RNA (dsRNA) viruses with three to seven dsRNA segments, comprising from 8.9 to 16 kb of genomic material [12]. They are pathogens of ascomycetous or basidiomycetous fungi, plants, and possibly insects. Interestingly, a chrysovirus was recently detected in a 1000-year-old ancient maize cob, as the oldest confirmed plant virus [14]. Viruses associated with this family are known to impinge on fungi, infecting humans and plants [7,8]. Recently, Konstantinidis et al. reported on the finding of an unclassified chrysoviral agent, in Greece, in *Uranotaenia unguiculata* and *Culex modestus* mosquitoes [13]. Outside the four protein coding segments and similarity to Hubei chryso-like virus 1, little is thus far known about its host spectrum or pathogenesis.

Serbia is known to be endemic for arthropod-borne pathogens [15]. A number of arboviruses is known to circulate in Serbia, e.g., Crimean–Congo hemorrhagic fever virus (CCHFV), West Nile virus (WNV), tickborne encephalitis (TBE) virus, and Usutu virus (USUV) [16,17,18]. Recently, a metagenomic study of arthropod viromes performed on arthropods collected from Belgrade, Serbia, yielded 21 viruses pertaining to 11 families, with 11 of them characterized as novel species [19]. Here, we report on a fortuitous discovery of an unclassified Xanthi chryso-like virus (XCLV) in Serbia, within a study targeting WNV detection.

## 2. Materials and Methods

### 2.1. Samples Collection and Study Area

The analyzed mosquitoes were collected within the scope of a routine WNV environmental screening program, regularly implemented by the Institute for Biocides and Medical Ecology, Belgrade, Serbia (IBME). Insect sampling was carried out on multiple sites in the north of Serbia during the period 2018 to 2020, concomitant to the time of the largest human epidemic of WNV infections in Southeast Europe up to that point (www.ecdc.europa.eu, accessed on 15 April 2022).

Field sample collection was performed using specialized BG Sentinel^®^ (Type 1 and Type 2) mosquito traps, which contain dry ice (CO_2_ in solid state, in the form of pellets) as an attractant, placed in a well-insulated container (volume 3 L). Upon setting the trap, CO_2_ is released, which attracts female mosquitoes. During trap collection, live mosquito specimens were separated from dead ones in situ. Living specimens were stored in a portable refrigerator for sample transport and were momentarily frozen on dry ice (−80 °C). Once frozen, the mosquitoes were counted, sorted, and transferred into corresponding labeled (date, location, city) sample tubes upon a preliminary identification at the genus level. These were stored on dry ice and taken back to the IBME where the mosquitoes were pooled in groups of approximately 50 individuals and subsequently homogenized prior to RNA extraction.

### 2.2. Nucleic Acid Extraction and qPCR

A total of 200 mosquito pools was submitted to nucleic acid extraction performed by Ribo-Sorb Nucleic acid extraction kit for the isolation and purification of RNA/DNA (REF K-2-1, Sacace Biotechnologies, Como, Italy, https://www.sacace.com/dna-rna-purification.htm, accessed on 15 April 2022). During the extraction and subsequent PCR, internal standards as well as positive and negative controls aimed to monitor the purification, amplification, and detection processes were used. Viral RNA was eluted in 30 µL of RNAse free H_2_O. Firstly, all isolated RNA was analyzed for the presence of WNV via real-time PCR kit for the qualitative detection of WNV RNA, according to the commercial PCR test protocol (V53-50FRT, Sacace Biotechnologies, Como, Italy, https://www.sacace.com/dangerous-infections.htm#s6, accessed on 15 April 2022). Extracted RNA of mosquito pools found positive by real-time PCR was used for further in-depth molecular and phylogenetic characterization.

### 2.3. Conventional PCR and DNA Sequencing

Initial PCR amplification was performed according to the nested PCR protocol for the WNV *NS5* gene. In the first round of amplification (outer PCR), isolated RNA was subjected to one-step RT PCR, with the degenerate primers covering partial WNV *NS5* gene: 20 pmol of 1NS5F and 1NS5R primer pair, in the total volume of 25 µL, according to the manufacturer’s instructions (Takara Bio Inc., Shiga, Japan, https://www.takarabio.com/products/mrna-and-cdna-synthesis/cdna-synthesis-kits/primescript-cdna-synthesis-kits/primescript-one-step-kit, accessed on 15 April 2022). The second PCR round (inner PCR) was conducted using the DreamTaq Hot Start PCR Master Mix (Thermo Fisher Scientific, Waltham, MA, USA), with 50 μL reaction mix contained 5 μL DNA product from the first-round reaction and 20 pmol each of sense 2NS5F and antisense 2NS5R primer. Of note, both rounds of nested PCR, included degenerate primers that were previously constructed and described for the *NS5* gene of WNV [20].

Conventional agarose gel electrophoresis was used to analyze all PCR products, initially revealing bands of two different length, 1019 bp and 832 bp.

In the process of detailed identification of the obtained amplified products, we designed XCLV-specific primers on the basis of the sole XCLV sequence existing in the public databases at the time of study (accession no. MW520404 [13]). These sets of primers (Xanthi_out_F, Xanthi_out_R, Xanthi_inn_F, Xanthi_inn_R) amplified a 735 bp product of the *P3* putative protease gene of XCLV.

Universal primers for the 16S mitochondrial ribosome subunit were used in confirmation of mosquito species along with virus-specific primers. Final confirmation of insect species was performed by PCR detection of 16S RNA (one-step RT-PCR Takara Bio Inc., Shiga, Japan), using universal 16S primers [21]. The list and description of all primers used in this research are given in Table 1.

The obtained PCR products were firstly purified with a MinElute Purification Kit (Qiagen, Hilden, Germany) kit, using the manufacturer’s instructions, and further directly sequenced using BigDye Version 3.0 Dye Terminator Cycle Sequencing Kit (Applied Biosystems, Foster City, CA, USA) on an ABI 3730 automated capillary sequencer. Results were analyzed with the Sequencing analysis software v.5.2 and assembled with SeqScape software v2.5 (Applied Biosystem, Waltham, MA, USA). Sequence identification was performed through NCBI nucleotide BLAST analysis (https://blast.ncbi.nlm.nih.gov/Blast.cgi, accessed on 15 April 2022).

All XCLV sequences generated during the course of this research were deposited in NCBI GenBank database.

### 2.4. Phylogenetic Analysis

Phylogenetic analysis was performed on the XCLV 735 bp *P3* gene sequences alongside genetically related chryso-like virus (CLV) sequences. Considering the fact that prior to this study a single *P3* putative protease gene XCLV sequence was available in the GenBank database (accessed October 2022), we also included CLV sequences of the corresponding genomic region, based on NCBI nucleotide BLAST search, optimized for somewhat similar sequences. Sequence alignment was performed by MAFFT 7 software (https://mafft.cbrc.jp/alignment/server/, accessed on 15 April 2022) and manually inspected (sequences included in the phylogenetic analysis are presented in the Appendix A). Prior to further phylogenetic analysis, alignment was screened for recombination using algorithms as implemented in the Recombination Detection Program 4 (RDP4) (http://web.cbio.uct.ac.za/~darren/rdp.html, accessed on 15 April 2022) [22]. The jModeltest 2.1.10 software statistically selected General Time Reversible (GTR) evolutionary model, with the gamma distributed rate across sites and proportion of invariable sites, as the best fitting substitution model using all 88 proposed models (https://mybiosoftware.com/jmodeltest-phylogenetic-model-averaging.html, accessed on 15 April 2022) [23]. The MEGA X software package was used to conduct phylogenetic analysis by maximum likelihood (ML) method selected and nucleotide substitution model with 1000 bootstrap replicates (https://www.megasoftware.net/, accessed on 15 April 2022) [24]. Phylogenetic tree topology was further confirmed by Bayesian phylogenetic analyses implemented in MrBayes software v3.2.7 (https://nbisweden.github.io/MrBayes/download.html, accessed on 15 April 2022) [25]. Phylogenetic trees were visualized using the FigTree program v 1.4.4 (http://tree.bio.ed.ac.uk/software/figtree/, accessed on 15 April 2022) [26].

## 3. Results

In 4/200 (2%) of studied mosquito pools, the obtained PCR product with WNV *NS5* gene-specific primers was shorter than expected: 832 bp instead of 1019 bp, as expected for WNV partial polymerase gene. BLAST analysis revealed the highest similarity of the obtained 832 bp product to XCLV P3 putative protease gene, with the single existing XCLV sequence in the GenBank database (accession no. MW520404). Of note, the results and analysis of the obtained WNV sequences are presented elsewhere (Todorovic et al., submitted). Identification of XCLV sequences was further confirmed with a separate, additional PCR amplification experiment by XCLV specific primers for 735 bp partial *P3* putative protease gene, where BLAST analysis of the obtained sequences again indicated the highest similarity to XCLV (accession no. MW520404) and Soufli chryso-like virus (accession no. MW520403) with 96.38% and 89.2%, respectively. The highest similarity at the protein level was observed with P3 putative protease of XCLV (accession no. QRD99897) and P3 putative protease of Soufli chryso-like virus (accession no. QRD99896), with 96.99% and 95.31%, respectively. The newly obtained XCLV sequences were submitted to the NCBI GenBank with accession nos. OP700407, OP700408, OP700409, and OM141591.

A topographical distribution of mosquito collection sites where XCLV was identified is shown in Figure 1. All three XCLV-positive sampling sites were found within suitable natural environments, providing favorable conditions for the survival of mosquitoes. All XCLV-positive mosquito pools were identified as *Cx. pipiens*, based on BLAST identification upon 16S RNA sequences.

XCLV-positive pools were collected in 2020 in the capital city of Belgrade and Sombor and Apatin municipalities in the Northern Serbian province of Vojvodina. The geographical position relative to the site of initial XCLV detection in Greece is depicted in Figure 1.

The final alignment for phylogenetic analysis comprised 15 viral sequences (5 XCLV and 10 CLV sequences) of the *P3 protease* gene, which is the total number of sequences of the relevant region available in the GenBank database at the time of analysis (accessed October 2022). Potential recombination events were not detected in the studied alignment. Average nucleotide distance of all analyzed nucleotide sequences was 0.55 (0.0–0.92, SD = 0.34), with amino acid divergence of 0.12%. Comparative analysis of XCLV sequences from Serbia revealed a nucleotide distance of 0.02, while the average nucleotide distance between XCLV sequences from Serbia and Greece was 0.07 (0.06–0.08).

The general topology of the phylogenetic tree for the studied dataset showed the existence of two separate clades with high bootstrap support, matching to the territory of the sequence origin (Figure 2). The first clade consisted of seven Hubei chryso-like virus sequences, of which five are from Australia and two from USA, collected in 2015 and 2017, respectively. The second tree clade included four XCLV sequences from Serbia (collected in 2020) together with XCLV and Soufli chryso-like virus from Greece (collected in 2020 and 2018), indicating the existence of evolutionary relatedness of the XCLV isolated from different countries from the Balkans. The tree was midpoint-rooted (Figure 2).

## 4. Discussion

Here, we report on a serendipitous detection of the uncommon XCLV associated with *Cx. pipiens* mosquitos collected in Serbia, Southern Europe. The finding is particular in the fact that it associates XCLV with a new potential carrier species and documents a novel geographical area of its distribution.

As vectors of infectious agents, mosquitoes are hematophagous arthropods of great public health importance. The investigation of mosquito viromes can yield a veritable flood of genomic data, with the focus often placed on agents of public health importance. We draw attention to the detection of a member of *Chrysoviridae* mycoviruses, a group of mosquito-borne viruses not known to infect humans, yet of tentative implications for the health of man, via their effect on fungi as primary hosts [8].

Up to now, in the literature, there was only a single report of XCLV detection described in an isolate from mainland Greece by Konstantinidis et al. [13] in *Uranotaenia unguiculata* (and possibly *Cx. modestus*) mosquitoes. Here, we present the detection of XCLV sequences in *Cx. pipiens*, implying a new potential arthropod host for this chrysovirus. The site of detection in the north of Serbia is both environmentally distinct and topographically distant (approximately 600 km) from the location of the primary XCLV findings.

In Serbia, the most comprehensive investigation of the viromes of multiple species of arthropods, sampled mostly in the capital city of Belgrade, was performed during 2016, based on Illumina MiSeq sequencing [19]. Captured arthropods represented nine species, and most of them were hematophagous, including mosquitoes, ticks, bedbugs, fleas, etc. Identified viruses represent a variety of RNA virus taxa with +ssRNA, −ssRNA, and dsRNA genome and are specific to invertebrates, vertebrates, or even to fungi and plants. Regarding mosquitoes, a great heterogeneity of viral sequences has been identified, such as those that fell within the families and order of *Bunyavirales*, *Picornavirales*, *Mononegavirales*, *Narnaviridae*, *Virgaviridae*, *Reoviridae*, and *Luteoviridae*. The research emphasizes the very common occurrence of some viral infections in the studied hematophagous arthropods captured in Serbia. However, identification of nine tentative novel viral species indicated the need for further molecular investigation of mosquitoes in order to detect and characterize new viral species [19].

Mosquitoes in this study were predominantly trapped in the northern and, to a lesser extent, eastern parts of Serbia. Three of the largest rivers in the country (Danube, Sava, and Tisa) pass through this area; moreover, large areas are covered by swampy terrain, which is mostly suitable for the breeding of mosquitoes. Considering the fact that some viruses (such as WNV) are maintained in an enzootic cycle [27] between birds as natural hosts and mosquitoes of the genus *Culex* (family *Culicidae*), as primary competent vectors, the diversity of birds may prove very important. Serbia is very rich in habitats and bird species, of which many are important for local and international conservation. Among them, Vojvodina’s nature reserves are some of the most studied and best known to the public. In this northern province of Serbia, from the 19th century until now, 312 species of birds have been recorded, which represents 42% of all bird species in Europe. In addition, 196 bird species nest in Vojvodina today, which represents 82% of Serbia’s birds (www.birdwatchserbia.com, accessed on 15 April 2022).

The *Chrysoviridae* family comprises two genera, *Alphachrysovirus* and *Betachrysovirus*, currently including 20 and 11 viral species, respectively, and with a growing number of unclassified, chryso-like viruses. XCLV has only recently been described, with a single report in the literature of metagenomic XCLV nucleic acid detection in mosquitos, by Konstantinidis et al. [13], namely, in *Uranotaenia unguiculata* and *Cx. modestus* mosquito species. In the last decade, the metagenomic approach to virome identification has greatly expanded our knowledge of viral diversity and placed arthropods in the central role driving viral evolution [28,29]. Here, we present the detection of XCLV sequences in *Cx. pipiens*, implying a newly described potential arthropod carrier for this chrysovirus. Notably, sole detection of viral genetic material does not allow to discriminate the true link between a vector and a carrier, that is, whether the mosquito is a biological or mechanical vector for XCLV, or the virus is merely “hitchhiking” the fungal, plant, or other associated material. Hence, we use the term “potential carrier” throughout to underscore this. It is worthwhile noting, however, that a wider areal presence/dispersion of a viral agent, all the while associated with an arthropod, may well insinuate at a vector–pathogen association.

The site of detection in the north of Serbia is both environmentally distinct and topographically distant (approximately 600 km) from the location of the primary XCLV finding in Xanthi, regional unit of East Macedonia and Thrace, northeastern Greece. Phylogenetic analysis of the included chryso-like viruses further confirmed BLAST identification of newly obtained XCLV sequences, based on highly supported clustering with Soufli chryso-like and Xanthi chryso-like viruses. General tree topology suggested the existence of clusters matching the viral species and territory of sequence origin. In this regard, one of the clusters was composed of Hubei chryso-like sequences originating from Australia and USA, and the other one contained Soufli chryso-like and Xanthi chryso-like virus sequences from Serbia and Greece, whereas the third cluster comprised Shuangao chryso-like viral sequences from Australia. Unfortunately, an in-depth phylogenetic analysis of XCLV sequences was not possible, given the small number of five available sequences in public databases. The full extent of the XCLV distribution, be it geographical or vectorial, is yet to be elucidated. The addition of new XCLV sequences will aid in mapping the presence of this virus.

Since the first definitive description of mycoviruses some 60 years ago, it has been recognized that these viruses are ubiquitous in all major groups of filamentous fungi [1]. *Chrysoviridae* are considered to be involved in impairing phenotypes of the rice blast fungus *Magnaporthe oryzae* [8], the most destructive pathogen of rice worldwide [9]. *Magnaporthe oryzae* chrysovirus 1 strains A (McCV1-A) and B (McCV1-B) were the first reported mycoviruses causing reduced pigmentation, changed colony morphology, and weakened growth, all signs of hypovirulence in the host fungus [9]. This trait may possibly be exploited to impart a useful effect on industrial plant life.

Intriguingly, chrysoviruses were found to impinge on human fungal pathogens as well. Notably, the product of the McCV1-A was suggested as a protein candidate for a pharmaceutical agent against *Cryptococcus neoformans* disease [7]. Moreover, the Alternaria alternata chrysovirus 1 (AaCV1) modulates the pathogenicity of its namesake fungal host, *A. alternata* [9]. *Alternaria* is known to cause opportunistic infections in immunocompromised humans [30], so the impaired growth caused by the AaCV1 might prove advantageous against the disease. The Aspergillus fumigatus chrysovirus (AfuCV) has also been identified in the eponymous *Aspergillus fumigatus* fungus [6]. Mycoviruses have also been suggested as epigenetic factors inducing alterations in the pathogenicity of plant-infecting fungi [9]. In recent research by Ejmal et al. [31] the authors observed the *Aspergillus thermomutatus* chrysovirus 1 to exert a reduction in the number of conidia from a clinical isolate of the titular fungal pathogen at 20 °C, suggesting the possible use of the virus as a biological control agent [11]. Finally, mycoviruses have been hypothesized as tentative therapeutic agents in fungal diseases of humans as well [10].

The genus Culex comprises more than 760 taxa, some of which are the most significant vectors of human diseases. Environmental changes influence their geography, heralding changes in their habitat. Our findings of the XCLV associated with the *Culex* arthropod imply their possible role as XCLV carriers, hinting at a larger distribution of mycoviruses in general throughout insect reservoirs and geographic areas alike. We speculate that, besides research oriented towards screening of arthropods for actual or potential human pathogens, an interesting alternative approach might be to search for arthropod-borne microorganisms with adverse effect on their other-than-human hosts (e.g., fungi) and thus of potential beneficial impact on human health and/or in industry. In this regard, it might be of relevance for future research to investigate if XCLV possesses mycopathogenic properties. Metagenomic approach has undoubtedly shifted the paradigm of viral research. Besides characterization of viromes of a number of diverse hematophagous and other arthropods, these methods, coupled with downstream phylogenetic and bioinformatics approach, allow to assess evolutionary and ecological links on a much larger scale [32,33,34]. Furthermore, our results underscore the limitations of Sanger sequencing of classical PCR products in fully depicting virome content and discerning mixed-virus infections, thus pointing to the relevance of the next-generation sequencing (NGS) approach. Namely, most molecular assays target only a limited number of already known pathogens using specific primers or probes, while metagenomic approaches characterize all DNA or RNA present in a sample, enabling the analysis of an entire genome. In recent decades, the application of the metagenomic approach together with bioinformatics has become a powerful tool for rapid detection, identification, and analysis of novel viruses, including those potentially infectious to humans. The viromes of some hematophagous arthropods have been well characterized using metagenomic methods [32,33], and lots of novel arthropod-specific and vertebrate viruses have been identified. Nevertheless, the Sanger sequencing approach still provides a valuable resource for monitoring and discovery of novel viral species, especially in smaller-scale studies and if used in concert with NGS methods.

We must concede to a certain possible drawback of our study. Namely, we did not perform testing regarding whether or not the XCLV sequence has been integrated into the genome of the Cx. pipiens mosquito. It is a well-known fact that viral sequences may be found embedded in arthropod genetic material [35,36]. We were unable to inquire for the presence of mosquito nucleic acids, as the samples we worked with were already RNA extracts of mosquito pools. Hence, as no DNA was available for detailed studies, this effectively precluded any possibility of testing for integrated viral DNA. In a previous work that analyzed the presence of arthropod-associated viral pathogens on the territory of Serbia, an NGS approach was utilized [19]. Herein, Stanojevic and coworkers identified 11 novel viral species. Moreover, the authors described chrysoviruses (mycopathogens) from the sequence dataset obtained from mosquitos; as in our work, again, the samples were not screened for integrated nucleic acids. Indeed, mosquitos have a considerable diversity of viruses, and the majority of these may not be associated with vertebrate organisms. Unlike in our study, the research by Shi and colleagues [37], which endeavored to characterize the total transcriptome of mosquito pools, elaborates on new viral species therein described as “unlikely to represent endogenous viral elements (EVEs)”. We wish to note that all of the sequences involved in our study are also present in the mentioned study [37]. Hence, the fact that we did not assay for this contingency may perhaps further obscure the role the mosquito may, or may not, play as a potential carrier. Although we did not perform PCR on DNA extract to exclude the possibility of these viruses being endogenous viral elements (EVEs), these viruses are not likely to be EVEs as they are present as complete genomes or CDSs without any interruptions. Additional in-depth molecular analysis of possible integration events by XCLV, and chrysoviruses in general, will elucidate this dilemma. However, our work supports the fact that the viromes of various hematophagous arthropods are still poorly understood, especially in countries such as Serbia, which is known to be an endemic region for arthropod-borne pathogens [15,19]. Moreover, virome analysis is also important in birds and other vertebrates, given the fact that these animals are part of the lifecycle of a number of arthropod-related viruses.

## 5. Conclusions

Our work broadens the arthropod-related horizon of XCLV circulation across ecosystems of the Balkans. The topographical/evolutionary divergence among the mosquito viromes calls for site-to-site field sampling and next-generation sequencing procedures in order to precisely map an exchange of viruses between species distributed over diverse ecologies and expand our understanding of potentially useful pathogens. This research will hopefully encourage a new approach to sequencing of the arthropod virome, with a special focus on microorganisms advantageous to humans and the environment.

## Figures and Tables

**Figure 1 tropicalmed-08-00225-f001:**
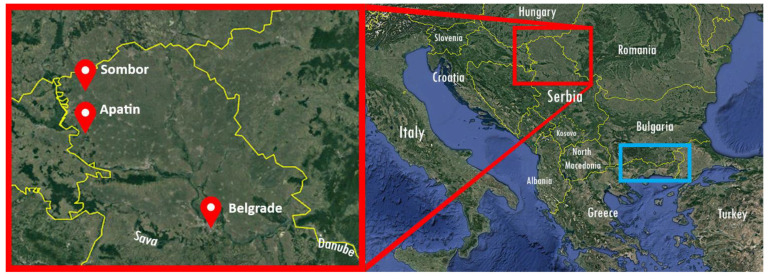
Map of the north of Serbia (**left**) with red pins indicating mosquito-sampling sites where the Xanthi chryso-like virus (XCLV) was detected (GPS coordinates of the sites are 44°85′07″ N, 20°46′97″ E; 45°76′29″ N, 19°08′47″ E; 45°66′58″ N, 18°97′98″ E). Area marked in blue (**right**) designates the geographic area where XCLV was first described by Konstantinidis and coworkers [13].

**Figure 2 tropicalmed-08-00225-f002:**
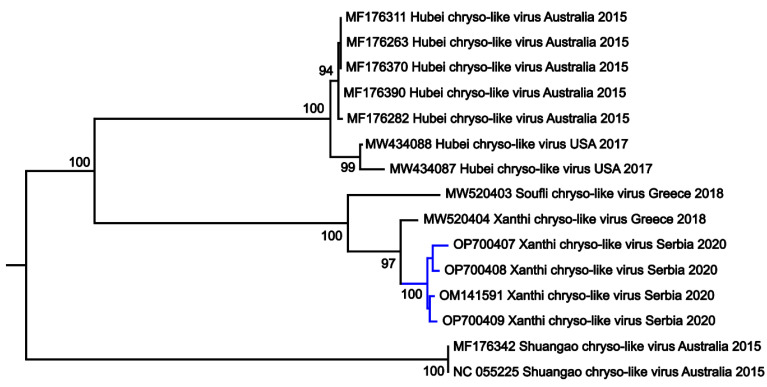
Maximum likelihood phylogenetic tree generated by MEGA X software, based on 735 nt of 15 chryso-like viruses. Bootstrap values > 65% are shown in the tree nodes. Sequences are indicated by GenBank accession number, name of the virus, country, and year of isolation. Newly detected sequences from Serbia are labeled in blue.

**Table 1 tropicalmed-08-00225-t001:** List of primers used in the study.

Target	Primer Name	Primer Sequence (5′-3′)	Annealing Temperature (°C)	Primer Position in Genome (nt)/Expected Product Length (bp)	Reference
16S rRNA	16 Sar	CGCCTGTTTATCAAAAACAT	60	837-1416/579	[21]
	16 SBR OiR	CCGGTCTGAACTCAGATCACGT			
WNV *NS5* gene	1NS5F	GCATCTAYAWCAYNATGGG	50	9035-10146	[20]
	1NS5R	CCANACNYNRTTCCANAC			
	2NS5F	GCNATNTGGTWYATGTGG	50	9103-10122/1019	
	2NS5R	TRTCTTCNGTNGTCATCC			
XCLV	Xanthi_out_F	TGCGGTGTGACAT	52	2108-2904	Designed in this study *
	Xanthi_out_R	AATATTACCAGCTT			
	Xanthi_inn_F	TTACTTGTGCAGGTACT	52	2141-2876/735	
	Xanthi_inn_R	GGGCAGATCTAATTCCA			

* The primer position on the 3rd segment of the XCLV genome was designed according to the sequence MW520404 from the NCBI database; WNV: West Nile virus; XCLV: Xanthi chryso-like virus.

## Data Availability

All sequenced data are submitted to the NCBI GenBank. All other data that support the findings of this study are in the manuscript or are available from the corresponding author upon reasonable request.

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
