# Peer review of "Detection of the Xanthi Chryso-like Virus in New Geographical Area and a Novel Arthropod Carrier"

_tropicalmed, 2023, doi:10.3390/tropicalmed8040225_

Round 1

Reviewer 1 Report (Previous Reviewer 1)

To the Editors of Tropical Medicine and Infectious Disease

The manuscript entitled “Detection of the Xanthi chryso-like virus in new geographical area and a novel potential arthropod vector.” It is interesting and, although simple, reports on a new arbovirus in Serbia. Therefore, I believe that it is of interest to the Special Issue to publish these results. I reviewed the first version of this manuscript and must say that the manuscript has improved significantly since then. The authors addressed all issues I raised in the first version of the manuscript. Therefore, I believe this manuscript is ready for publication.

Author Response

As the Reviewer 1 suggests publication without any changes, we have no further comments and thank the Reviewer 1 for their time and effort into reviewing our manuscript.

Reviewer 2 Report (Previous Reviewer 2)

Dear authors,

Thanks for your respones to my comments. The problem of what I say is that you should provid the strong evidence to prove the detections. PCR and Sanger sequencing are very sensitive and are the ambiguous evidence. As you say, the PCR products were come from the detection of WNV, but resulted the sequences of Xanthi chryso like virus. It is also unclear, although you have made the gel purification. What about northern blot and virus particle observation.

Author Response

Comment: Thanks for your respones to my comments. The problem of what I say is that you should provid the strong evidence to prove the detections. PCR and Sanger sequencing are very sensitive and are the ambiguous evidence. As you say, the PCR products were come from the detection of WNV, but resulted the sequences of Xanthi chryso like virus. It is also unclear, although you have made the gel purification. What about northern blot and virus particle observation.

Authors’ answer: We thank the Reviewer 2 for their suggestions. However, we must strongly disagree that Sanger sequencing as well as PCR are ambiguous. Namely, we have made a fortuitous discovery of the XCLV via WNV primers, as explained; afterwards, we designed our own XCLV-directed primers, which successfully amplified the nucleic acid of this virus. Furthermore, we verified these findings via Sanger sequencing, and the results were BLAST-confirmed subsequently. We strongly believe that this is enough of an identification approach, as Sanger sequencing can be used even after next generation sequencing for the verification of microorganisms.

We have not used Northern blot, as PCR and Sanger sequencing, if used in concert as we did, are more sensitive.

Virus particle observation would not be of much help, as it can discern only to the level of the family, and cannot differentiate between types of chrysoviruses. Even if it would have been possible to identify a chrysovirus by electron microscopy, we would still have to perform PCR and Sanger to verify. Also, we had only RNA extracts of mosquito pools to work with, and all of the virions would be destroyed during the process of nucleic acid extraction. At the very last, this is only the second instance of XCLV being described in literature so far, and a more comprehensive analysis surpasses by far the scope and aim of our investigation.

We are deeply convinced that our methodology is sound, clearly explained and elaborated in sufficient detail.

Reviewer 3 Report (New Reviewer)

Please see comments in the pdf file.

Author Response

Please see the attachment for a point-by-point response.

Round 2

Reviewer 2 Report (Previous Reviewer 2)

The problems in the manuscript remains what I mentioned earlier.

Author Response

We kindly thank very much the Reviewer 2 for their effort in reviewing our Manuscript. We greatly appreciate the time and energy they invested and hope that their suggestions have made our work a better one. We must, however, stand by our previous comments and remain firm in the opinion that we have fully addressed all of the issues in our Manuscript.

Reviewer 3 Report (New Reviewer)

 Please clarify in teh text why your detected genoma is not in fact viromic genoma integrated in mosquito genoma.

Author Response

Many thanks to the Reviewer 3 for their effort and time invested in reviewing our work. The viral genome integration is, indeed, an intriguing supposition. As stated previously, we have not, in fact, checked whether this integration has occurred. Again, as far as we are aware, there is no for this virus being able to integrate into the genome. However, this should be included in the limitation part of the study, and we thank again the Reviewer 3 for raising this question, as it will no doubt help to further improve our research. We would suggest including the following paragraph into the text, and present it here for the Reviewer 3’s consideration:

(lines 346 to 369): “We must concede to a certain possible drawback of our study. Namely, we have not performed testing on whether or not the XCLV sequence has been integrated into the genome of the Cx. pipiens mosquito. It is a well-known fact that viral sequences may be found embedded in arthropod genetic material [34,35]. We were unable to inquire for the presence of mosquito nucleic acids as the samples we worked with were already RNA extracts of mosquito pools. Hence, as no DNA was available for detailed studies, this effectively precluded any possibility of testing for integrated viral DNA. In a previous work that analyzed the presence of arthropod-associated viral pathogens on the territory of Serbia, an NGS approach was utilized [19]. Herein, Stanojevic and coworkers have identified 11 novel viral species. Moreover, the authors describe chrysoviruses (mycopathogens) from the sequence dataset obtained from mosquitos; as in our work, again, the samples were not screened for integrated nucleic acids. Indeed, mosquitos have a considerable diversity of viruses, and the majority of these may not be associated with vertebrate organisms. Unlike in our study, the research by Shi and colleagues [36] which endeavored to characterize the total transcriptome of mosquito pools elaborates on new viral species therein described as “unlikely to represent endogenous viral elements (EVEs)”. We wish to note that all of the sequences involved in our study are also present in the mentioned study [36]. Hence, the fact that we did not assay for this contingency may perhaps further obscure the role the mosquito may, or may not play as a potential carrier. Although we did not perform PCR on DNA extract to exclude the possibility of these viruses being endogenous viral elements (EVEs), these viruses are not likely to be EVEs as they are present as complete genomes or CDSs without any interruptions. Additional in-depth molecular analysis of possible integra-tion events by XCLV, and chrysoviruses in general, will elucidate this dilemma.”

(lines 488 to 495):

  1. Suzuki Y, Baidaliuk A, Miesen P, Frangeul L, Crist AB, Merkling SH, Fontaine A, Lequime S, Moltini-Conclois I, Blanc H, van Rij RP, Lambrechts L, Saleh MC. Non-retroviral Endogenous Viral Element Limits Cognate Virus Replication in Aedes aegypti Ovaries. Curr Biol. 2020;30(18):3495-3506.e6. doi: 10.1016/j.cub.2020.06.057.
  2. Gilbert C, Belliardo C. The diversity of endogenous viral elements in insects. Curr Opin Insect Sci. 2022;49:48-55. doi: 10.1016/j.cois.2021.11.007.
  3. Shi M, Neville P, Nicholson J, Eden JS, Imrie A, Holmes EC. High-Resolution Metatranscriptomics Reveals the Eco-logical Dynamics of Mosquito-Associated RNA Viruses in Western Australia. J Virol. 2017;91(17):e00680-17. doi: 10.1128/JVI.00680-17.

This manuscript is a resubmission of an earlier submission. The following is a list of the peer review reports and author responses from that submission.

Round 1

Reviewer 1 Report

Dear,

I’ve read and reviewed the manuscript of Jankovic and colleagues entitled “Broadening the Chrysoviridae horizons: a new geographical 2 area and potential arthropod vector for the Xanthi chryso-like 3 virus”. The manuscript is well-written, and clear, and raises the knowledge of the occurrence of new arboviruses (in this case a mycopathogenic virus) possibly transmitted by mosquitoes. The authors identified the Xanthi chryso-like virus (XCLV) in urban populations of Culex pipens by PCR and sequencing and performed an evolutionary analysis to better understand the relationship of this new virus with other viruses from the Chrysoviridae family. The manuscript brings novel findings and is on the scope of the special edition. I have only a few comments and questions and, therefore, I recommend the paper for publication after a minor review. Please find below my comments

1. In the introduction (lines 43-44) the authors suggest that mycopathogenic viruses may be favourable to humans. The statement is very vague and lacks references for better understanding. Later, in the Discussion, the authors give more information on this subject, providing examples and references. I suggest the authors be more specific on this subject in the introduction, rather than the discussion.

2. In line 60, the authors state that this is the second study to report on XCLV virus. However, there is no mention, in the introduction, on the first study (in Greece). Please provide the information and reference for the first study.

3. The PCR section (2.2) of the Material and Methods is confusing, and I had to read it many times to understand it. Here are my observations and questions on this section:

3.1. In lines 98-99 the authors specify that positive pools were transferred to Belgrade Facutly of Medicine, implying that the qPCR tests were performed somewhere else. This brings the question of where the qPCR was made and why different laboratories. I suggest avoiding this whole part for conciseness unless the experiments performed in Belgrade were paid as a service (which, if the case, should be mentioned).

3.2. In lines 100-109 the authors explain the PCR, but explain only the WNV NS5 primers. It is written that PCR was performed with degenerate primers followed by nested PCR. However, in table 1 XCLV primers do not degenerate.  Also, only in later sections becomes clear that the XCLV primers also target the NS5 gene. I must ask the authors to rewrite this section, so the text gets more clearer.

4. Line 118: the authors wrote: “… showing a single band on gel electrophoresis, of both expected lengths (1019bp)…”. I’m confused about how a single band can have two different lengths of the same size (1019). Please clarify.

5. Figure 1. The figure is nice and clear. However, I’m curious about all other collection sites not positive to XCLV. I understand that the authors wanted to keep the figure simple, clear, and focused on the results. However, it would be great to know how many populations were sampled between positive sites. Maybe, if the figure does not get too polluted by information, a supplementary figure showing all collection sites would be nice.

6. Line 267. Alternaria alternata is not in italic

Reviewer 2 Report

This study used PCR and Sanger sequencing in order to investigate possible WNV presence in a population of mosquitoes, but with a serendipitous detection of the uncommon XCLV. However, viruses in Chrysoviridae family, are commonly found in fungi and plants.

1.     Only a PCR test may not tell the XCLV is really infecting mosquito. For example, mosquitoes easily contact with some fungi

2.     The part of “Nucleic acid extraction and PCR” in “Materials and Methods” are describing the detection of WNV. But there are not results about the detection of WNV.

Reviewer 3 Report

Previously, a chryso-like virus termed Xanthi chryso-like virus (XCLV) was detected by a metagenome analysis of a mosquito species Uranotaenia unguiculata in Greece, whose genomic sequence has been deposited with the public databases but appears not to have been published in a journal. The current manuscript ID: tropicalmed-2104113 by Jankovic et al. describes the second example of detection of XCLV possibly in another mosquito species Culex pipiens. The authors detected the virus isolates in mosquitos collected in three different localities in Serbia by RT-PCR with a few primer sets designed based on the available sequence of XCLV dsRNA3. The amplified fragments were sequenced and then used to compare phylogenetically.

The manuscript carries nothing exciting and fails to address key virological questions and needs an overhaul with the text. Only novelty of this paper is the XCLV detection from C. pipiens, unreported as a host to XCLV, in Serbia where no XCLV was reported before. This reviewer is very hesitant to recommend this paper to be accepted, because of the paucity of novelty and poor writing as shown below.

Major points:

The paper contains grammatical errors, and some sentences lack accuracy and clarity. Some are shown below:

Title: Too flowery for the finding of the study.

Abstract: The authors are talking about arboviruse as human pathogens, and then suddenly mention mycoviruses in line 15. The abstract should provide important findings carried in the paper concretely and succinctly. However, the current abstract fails to provide such information on Xanthi chryso-like virus (XCLV). 

Introduction: As it stands, the introduction is not playing an expected role. The paper seems to report a second example of the detection of Xanthi chryso-like virus. However, no mention of the first example is made. Information on its provenance, segment number, host organisms, and similarity to plant/fungal chrysoviruses should be described.

Lines 48-50. Ref 8 does not show that the chrysovirus Magnaporthe oryzae chrysovirus 1 strain A (MoCV1-A) affects a human pathogenic fungus Cryptococcus neoformans. They showed ORF4 expression to be able to alter the growth and physiology of C. neoformans.

The authors describe arboviruses in the Abstract, Introduction and Discussion and highlight the potential of the virus to reduce virulence of pathogenic fungi to human and plant, without knowing whether XCLV can infect organisms other than mosquitos. It is likely that XCLV is a insect-specific virus.

L64. Why mycopathogenic viruses? The authors do not know it until it is substantiated.